# A Forest Fire Susceptibility Modeling Approach Based on Integration Machine Learning Algorithm

Changjiang Shi and Fuquan Zhang *

College of Information Science and Technology, Nanjing Forestry University, Nanjing 210037, China; 1213676009@njfu.edu.cn
* Correspondence: zfq@njfu.edu.cn

**Abstract:** The subjective and empirical setting of hyperparameters in the random forest (RF) model may lead to decreased model performance. To address this, our study applies the particle swarm optimization (PSO) algorithm to select the optimal parameters of the RF model, with the goal of enhancing model performance. We employ the optimized ensemble model (PSO-RF) to create a fire risk map for Jiushan National Forest Park in Anhui Province, China, thereby filling the research gap in this region's forest fire studies. Based on collinearity tests and previous research results, we selected eight fire driving factors, including topography, climate, human activities, and vegetation for modeling. Additionally, we compare the logistic regression (LR), support vector machine (SVM), and RF models. Lastly, we select the optimal model to evaluate feature importance and generate the fire risk map. Model evaluation results demonstrate that the PSO-RF model performs best (AUC = 0.908), followed by RF (0.877), SVM (0.876), and LR (0.846). In the fire risk map created by the PSO-RF model, 70.73% of the area belongs to the normal management zone, while 15.23% is classified as a fire alert zone. The feature importance analysis of the PSO-RF model reveals that the NDVI is the key fire driving factor in this study area. Through utilizing the PSO algorithm to optimize the RF model, we have addressed the subjective and empirical problems of the RF model hyperparameter setting, thereby enhancing the model's accuracy and generalization ability.

**Keywords:** forest fire prediction; PSO-RF; fire risk map; machine learning

## 1. Introduction

Forests are of utmost importance to our planet as they serve as critical ecosystems, playing essential roles in soil and water conservation, carbon cycling, and biodiversity preservation [1]. Forest fires can seriously damage forest ecology. The harm caused by forest fires is profound, with significant impacts on the natural ecology, economy, and society [2,3]. A vast number of forest fires occur worldwide every year, making them a hot topic in environmental protection [4]. The rise in global temperatures and the escalating occurrence of extreme weather events have contributed to a notable surge in forest fires across the globe [5]. The prominence of forest fires is growing, underscoring the practical and strategic significance of studying them. As of 2021, the forested area of Chuzhou City in Anhui Province, China totals up to 741.5 million acres, earning it the title of "National Forest City". As urbanization progresses and human activities become inevitable, the challenges associated with forest ecological management in Chuzhou City have become urgent. This paper selects Jiushan National Forest Park in the city as the research site.

The prediction of forest fires is of great importance [6]. The initial methods of forest fire prediction mainly relied on field observations, followed by the use of statistical analysis methods to identify potential fire sources [7]. This approach typically requires a significant amount of manual observation and data collection, which is tedious, time-consuming, and lacks accuracy [8]. The application of satellite remote sensing technology has transformed this situation. Researchers can collect, store, and analyze a large number of geospatial data

via satellites. This information serves as the foundation for devising intricate models to forecast forest fires. Early forest fire prediction models primarily based on general linear models proved inadequate in capturing the complex non-linear relationships between forest fire occurrence and environmental factors, leading to unsatisfactory results [9,10]. Subsequently, generalized linear regression models such as LR and Poisson regression have been widely applied in forest fire prediction models [11–13]. To some extent, these generalized linear models solve the issue of nonlinearity between forest fire factors.

During recent years, the utilization of machine learning algorithms has enabled researchers to develop effective and precise forest-fire prediction models. These methods have proven to be efficient and accurate in forecasting forest fire occurrences. Machine learning methods can use a large number of data for prediction. Supervised learning methods can use historical data to predict the probability of fire outbreaks and use classification or regression algorithms for data analysis [14]. Unsupervised learning methods can cluster data to identify fire outbreak patterns [15]. Reinforcement learning methods can optimize fire prediction models by learning how to take action [16]. Machine learning methods can consider multiple factors, including weather, vegetation, terrain, and human activities, and can handle a large number of data and complex models [17]. Among them, LR, SVM, and RF are classic machine learning models. The LR model is a classic classification algorithm, a linear classifier suitable for binary problems of whether a forest fire occurs [18]. However, the LR model is sensitive to outliers and cannot handle non-linear problems. If many data features are processed, it can lead to the curse of dimensionality. The SVM model is suitable for both binary and multi-class problems in classification. By using kernel functions, the SVM model maps data into a high-dimensional space and solves non-linear classification problems well, handling datasets with many features. However, the SVM model is sensitive to noise data and parameter choices, requiring data preprocessing [19]. The RF model randomly draws samples, generates multiple decision trees, and combines them through averaging or voting, reducing overfitting issues and improving the model's prediction accuracy [20]. The RF model reveals the non-linear relationships between various factors well; hence, it is widely applied in the field of forest fire prediction. However, because each algorithm has some defects, the accuracy of the model is affected. Faced with this situation, many researchers have found that the accuracy of ensemble models is superior to single models, so various types of ensemble models are extensively used in related fields [21].

Truong, in his study of landslide susceptibility probability, combined the bagging ensemble (BE) and logistic model trees (LMTree) to construct a BE-LMTree model. The results from the validation set showed a predictive accuracy of 83.4%, surpassing that of a single SVM model [22]. Lingxiao Xie, during his investigation into the susceptibility mapping of forest fires in Liangshan Prefecture, China, initially processed the triggering factors using the frequency ratio (FR) method. Subsequently, he utilized the Bayesian optimization (BO) algorithm to optimize the parameters of the XGBoost model, thus proposing the FR-BO-XGBoost model. Compared to the RF and SVM models, the FR-BO-XGBoost model demonstrated superior performance, with an AUC score of 0.887 [23]. Meriame proposed the frequency ratio–random forest (FR-RF) model (AUC = 0.858), which first processed the dataset with FR and then combined with RF to map the fire risk of forests in the Mediterranean region [21]. Obviously, in the field of machine learning models, the performance of ensemble models is higher than that of single models, and ensemble models are a current trend in the field of forest fire prediction. Among them, most ensemble models aim at optimizing the hyperparameters of machine learning models to achieve better models [21,23,24].

To achieve better models, hyperparameters in machine learning algorithms require manual setting and optimization. Traditional hyperparameter tuning methods often rely on expert experience, unwritten heuristics, or occasionally brute-force search techniques [24]. Obviously, this subjective method of parameter tuning results in unstable model accuracy. Certain researchers have attempted to tune models in a more systematic and reasoned manner. Bergstra used a random search method to tune the parameters of neural networks

and found that random search in the same domain can find equivalent or better models in a shorter computation time [25]. Snoek utilized Bayesian optimization with Gaussian process sampling to model the generalization performance of learning algorithms, with the objective of identifying the optimal parameter configuration for machine learning algorithms [24].

Hyperparameters in the RF model, such as the number of decision trees (n_estimators) and the maximum tree depth (max_depth), greatly affect the convergence speed and prediction performance of the RF model. Most researchers use default parameters in the RF model ensemble method or select an approximate parameter based on experience, but these hyperparameter selection methods may reduce the performance of the RF model. To obtain a better forest fire risk assessment model, this paper proposes an ensemble model (PSO-RF) that employs PSO to optimize key parameters within RF, thus avoiding the limitations and defects of traditional RF model parameter optimization methods [26]. To validate its performance, this paper also cites the LR, SVM, and RF models for comparative analysis. To gain further insights into the factors influencing wildfires, the optimal model was employed to evaluate the importance of different features. This analysis helped identify the primary triggering factors associated with wildfires in the region. At the same time, prediction accuracy, recall rate, F value, and AUC curve were utilized to assess the precision of the four models [27]. This paper fills the gap in forest fire research in the Jiushan area by conducting a study on forest fire risk assessment and provides a novel forest fire risk assessment method based on the PSO-RF ensemble model. The advantage of this method in terms of accuracy and generalizability has been demonstrated through comparison with traditional methods. The forest fire risk map of the Jiushan area, drawn based on the PSO-RF model, demonstrates superior accuracy compared to other models. This study's findings enhance forest stewardship in the Jiushan region, offering valuable insights to administrators for strategizing forest fire mitigation. Future research can further explore the implementation of other optimization algorithms in forest fire risk-assessment models, with the aim to provide more effective forest fire control and resource management solutions in a wider range of areas and scenarios.

## 2. Materials and Methods

### 2.1. Study Area

Jiushan is located within the jurisdiction of Chuzhou City, Anhui Province, located within the southern mountainous area of Fengyang County. It lies between 117°19′–117°48′ east longitude and 32°37′–32°46′ north latitude. The terrain is primarily hilly and relatively flat, covering an area of 249.12 square kilometers. Figure 1 shows the satellite image of the Jiushan area. Jiushan is located in a region characterized by a North subtropical monsoon climate. It maintains an average annual temperature of 14.9 °C and receives an average annual rainfall of 876 mm. During the developed monsoon period, the hottest month in Jiushan experiences an average temperature above 22 °C, while the coldest month typically sees temperatures ranging between 0–5 °C. Jiushan is situated in the transition zone of the north and south plant regions. The forest structure is mainly composed of deciduous broad-leaved forests of the Fagaceae family and evergreen conifers of the Pinaceae family. The famous tourist attraction, Jiushan National Forest Park, is located here [28]. The fire prevention period in Jiushan is from November each year to April of the following year.

### 2.2. Data Sources

Forest fire-related data can be divided into independent variable data and dependent variable data [29]. Independent variable data include slope, aspect, topographic wetness index (TWI), altitude, distance to roads (DTR), distance to population centers (DTP), normalized vegetation index (NDVI), and mean monthly temperature (MMT) [30]. The dependent variable data are the fire point. The slope, aspect, TWI, and altitude data are derived from the SRTMDEMUTM90M resolution digital elevation data in the DEM digital elevation dataset of the Geospatial Data Cloud platform, and the selected timeframe is March 2019. The distances between fire points and roads, as well as residential areas,

were obtained from the 2017 data in the 1:250,000 National Basic Geographic Database available in the National Geographic Information Resources Catalogue Service System. The normalized vegetation index and average monthly temperature data are sourced from Landsat-8 OLI (30 m resolution) satellite data [31], with the NDVI data coming from satellite images from March 2019 and the fire point data derived from Sentinel-2 satellite data (60 m resolution). All these data were processed using ENVI 5.3 and ArcMap 10.2. Since the resolution of satellite images is not uniform, we used ArcMap 10.2 software to resample the images based on the nearest neighbor allocation method. Table 1 shows the specific sources of the data, and considering subsequent fire point extraction, the grid resolution of all data is set to 30 × 30.

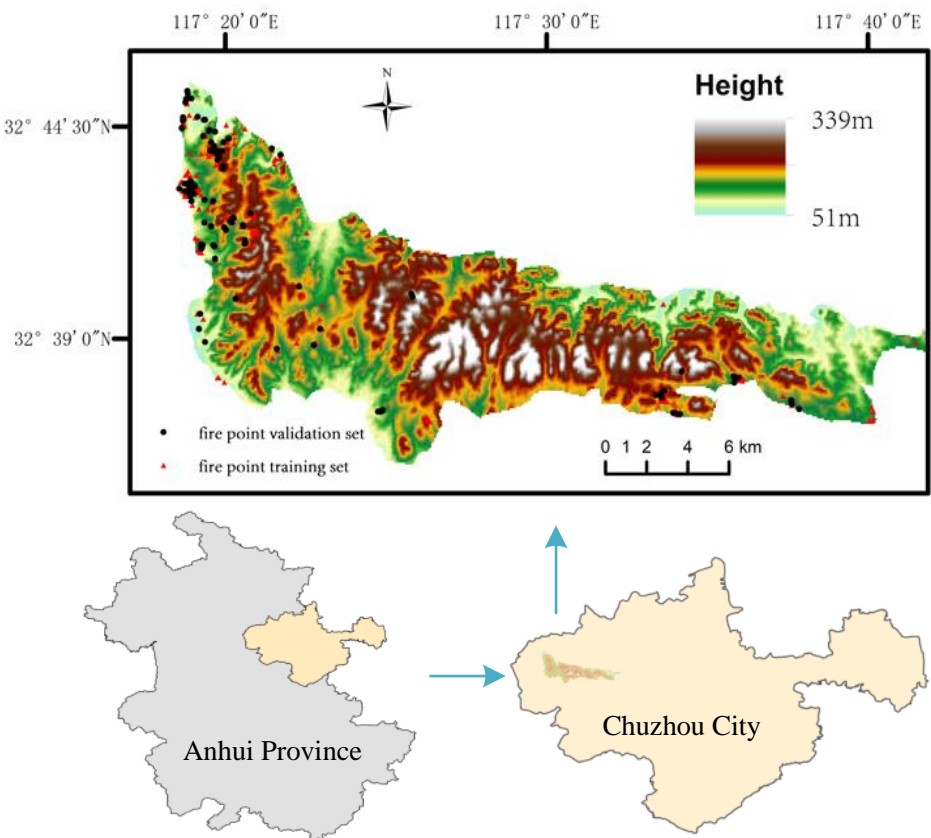

**Figure 1.** Study area.

**Table 1.** Classification of fire factors.

| Data Type | Factors | Data Range | | Data Source |
|---|---|---|---|---|
| | | Min | Max | |
| Terrain factor | Slope | 0 | 39.5 | https://www.gscloud.cn, accessed on 10 July 2021 |
| | Aspect | 0 | 360 | |
| | TWI | 3.29 | 24.6 | |
| | Altitude | 51 | 339 | |
| Human activity | DTR | 400 | 1500 | https://www.webmap.cn, accessed on 10 July 2021 |
| | DTP | 500 | 5500 | |
| Vegetation | NDVI | −0.64 | 0.58 | https://earthexplorer.usgs.gov, accessed on 10 July 2021 |
| Meteorological | MMT | 12.57 | 29.96 | |

### 2.2.1. Fire Point Data

In light of the inadequate availability of actual fire record data in this research, we collected data from the thermal infrared sensors on satellites using remote sensing algo-

rithms, which processed the infrared and near-infrared band data as historical fire data for the study area. We downloaded Sentinel-2 data with a time resolution of 10 days and a spatial resolution of 60m from the Copernicus Data Centre (https://scihub.copernicus.eu, accessed on 15 July 2021) and extracted a total of 396 fire points from 2018–2020 through band calculations. To match the spatial resolution of other data, we used the SNAP6.0 software from the European Space Agency as a preprocessing tool to resample the downloaded data to $30 \times 30$ m grids. Next, we extracted the near-infrared (NIR) and short-wave infrared (SWIR) bands and carried out band computations using the normalized burn ratio (NBR) index [32] and the differenced normalized burn ratio (dNBR) index [33].

$$NBR = (NIR - SWIR)/(NIR + SWIR) \tag{1}$$

$$dNBR = NBR_{pre-fire} - NBR_{post-fire} \tag{2}$$

The NBR is a vegetation index constructed from the NIR and SWIR bands. It leverages the spectral reflectance characteristics of burned areas, which increases in the SWIR band and decreases in the NIR band, thereby separating burned areas from other features. Based on the pre-fire and post-fire images' NBR, we computed the dNBR. As per the recommendations of the United States Geological Survey (USGS) [34], we used a dNBR threshold of 0.1 to delineate burned (>0.1) and unburned (<0.1) areas. Eventually, we extracted a total of 396 fire points from the burned areas based on the dNBR values for the years 2018–2020. Previous studies have shown that the accuracy of fire point extraction from Sentinel-2 images using the dNBR method can be as high as 86% [35,36], so the fire points extracted in this study can be used for fire prediction.

### 2.2.2. Terrain Factors

Terrain factors include altitude, slope, aspect, and TWI. As the altitude increases, the temperature tends to decrease while humidity levels tend to increase, which lowers vegetation dryness and flammability, thereby reducing the likelihood of forest fires. Higher-altitude areas are easily affected by precipitation, which increases vegetation moisture and lowers fire risk. The greater the slope, the longer the sunshine duration, which raises the temperature, accelerates surface water evaporation, and renders vegetation flammable. The TWI is a terrain factor that considers rainfall, soil water content, and vegetation coverage, reflecting the distribution and changes in surface water [37]. Areas with high TWIs usually have higher soil water content and vegetation coverage and hence better vegetation growth and lower fire risk levels. Conversely, areas with low TWIs are usually dry or semi-arid regions with poor soil water content and vegetation coverage, which are more prone to fires.

### 2.2.3. Vegetation Factors

The selected vegetation parameter is the NDVI, a powerful tool for observing the plant habitat. As a form of remote sensing information, NDVI demonstrates the geographical arrangement and abundance of greenery [38]. It represents the quotient obtained from the distinction between near-infrared red (NIR) and red (R) bands over their aggregate, yielding a range for NDVI between $-1$ and 1. Generally, clouds and water exhibit negative NDVI values, rocks and barren land register an NDVI of 0, and regions with more compact vegetation tend to have NDVI values nearing 1.

### 2.2.4. Human Activity Factors

Elements of human activity, such as roads and villages in Chuzhou City, were selected to illustrate the impact of human behavior on wildfire occurrences. As a result of urbanization, the population in the forested areas around cities is increasing, and a large number of roads have been built. Road construction intensifies the burning of forests for land clearance along the route, raising the likelihood of wildfires [39]. Villages are located in mountainous areas; on the one hand, the closer it is to a village, the more likely a wildfire

is to be triggered by human activities; on the other hand, wildfires in areas far from villages are often hard to detect and control in a timely manner, leading to an escalation of the fire.

### 2.2.5. Meteorological Factors

Temperature indices, which are representative in forest fire research, were chosen as meteorological factors. The mean monthly temperature at the time of fire point occurrence in the study area was obtained using the atmospheric correction method. As the temperature rises, the evaporation speed of vegetation water in forest areas will accelerate, and the dryness of vegetation will gradually increase. This, in turn, will increase the speed and range of fire spread, thus escalating the likelihood of forest fires [40]. Figure 2 shows the specific influencing factors.

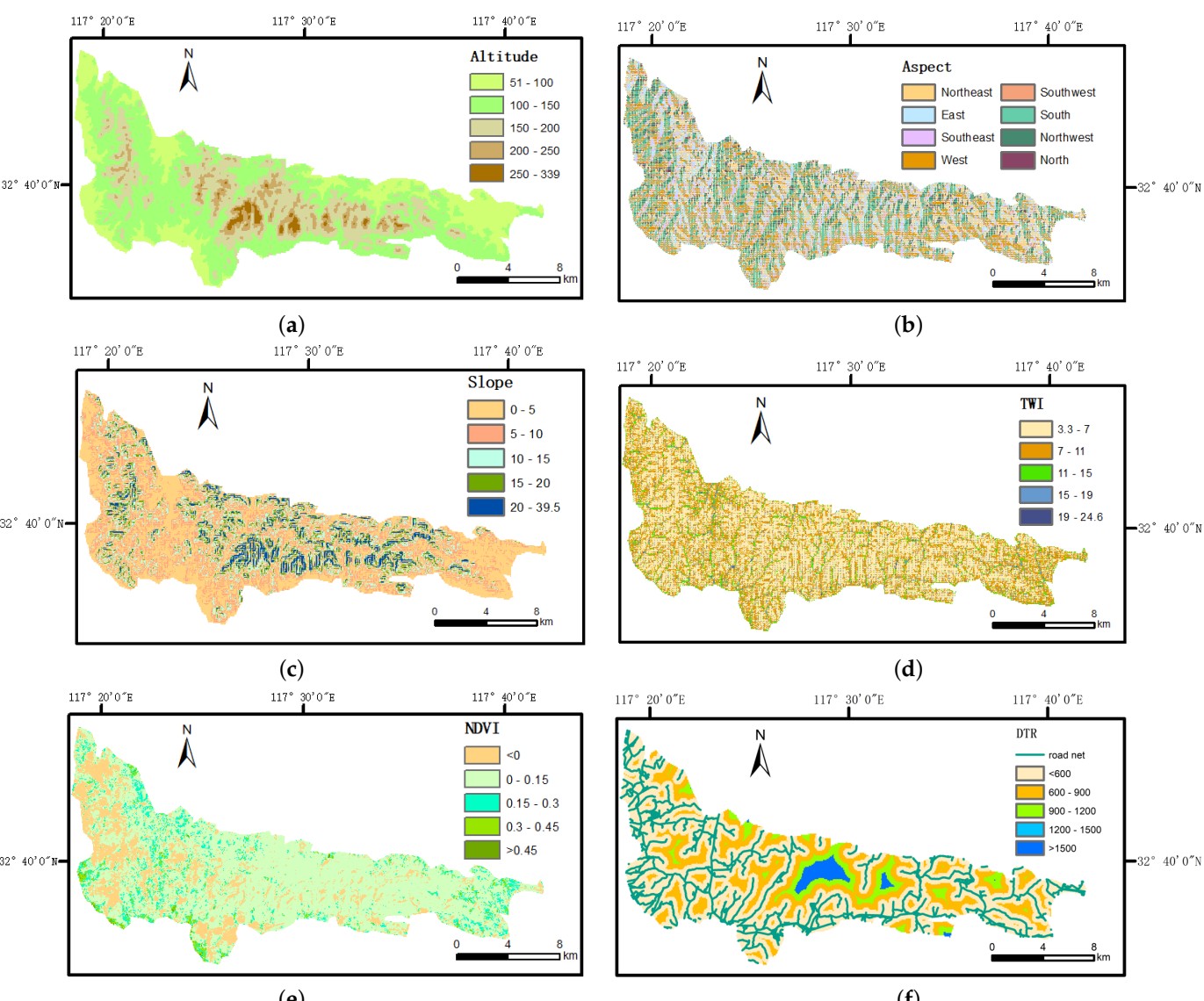

**Figure 2.** *Cont.*

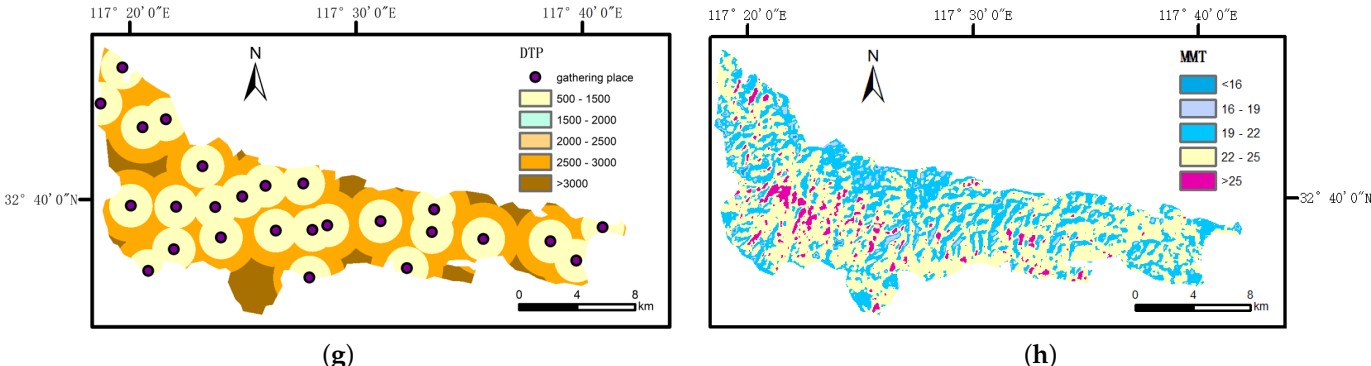

(**g**)  (**h**)

**Figure 2.** Distribution of fire impact factors. (**a**) Altitude, (**b**) aspect, (**c**) slope, (**d**) TWI, (**e**) NDVI, (**f**) distance to roads (DTR), (**g**) distance to population gathering point (DTP), and (**h**) mean monthly temperature (MMT).

### 2.3. Research Method

In this study, we selected the Jiushan area as our research site and identified a total of 396 fire points from 2018 to 2020. Data balance and randomness can enhance the predictive capacity of the model and align with the random application scenarios of future models. Therefore, based on spatial and temporal randomness, we selected 396 non-fire points, resulting in a total of 792 sample data points. Each sample contains relevant information about the point, and we split the samples into training and test sets in a 7:3 ratio [29,41,42]. We trained on four types of machine learning models using the training set and evaluated the performance of each model with the help of the validation set. Finally, we visualized the model results and drew a fire risk map of the research area [43,44]. The detailed process is shown in Figure 3.

#### 2.3.1. Dataset Configuration

This study uses machine learning models for classification prediction, so the dependent variable is divided into two categories: the occurrence and non-occurrence of forest fires. This study extracted a total of 396 fire points using remote sensing images. These fire points were randomly grouped, with 70% used as the training set and 30% as the validation set. The sample data also require non-fire point data. Based on previous experience, we adjusted the ratio of fire points to non-fire points to 1:1 [19,20].

#### 2.3.2. PSO-RF Model

Particle swarm optimization random forest (PSO-RF) is a hybrid approach that combines the PSO algorithm with the RF algorithm to tackle classification and prediction tasks. PSO-RF aims to enhance prediction accuracy and stability by optimizing RF model parameters using the PSO algorithm. Figure 4 shows the flowchart of the model.

The PSO model has the capacity for global search and rapid convergence, enabling it to find the optimal solution in a relatively short period of time. This avoids the trial-and-error and experience accumulation process of traditional optimization methods, thereby improving the efficiency of optimization. Simultaneously, the PSO model has a degree of adaptability and randomness, which helps avoid local optimal solutions and overfitting issues, thus optimizing the RF model more effectively [45].

The hyperparameters of the RF model, such as the number of decision trees (n_estimators) and the maximum depth of the tree (max_depth), significantly impact the model's convergence speed and predictive performance [46]. With a low number of decision trees, the model may experience underfitting. While a high number of decision trees can improve model performance, it also increases computation time and offers limited performance improvement. Similarly, appropriately setting the maximum depth of the tree is also crucial. Traditional parameter optimization methods are usually based on experience or trial and error; they are less efficient and are often unable to achieve the global optimal solution.

This paper uses PSO to optimize the parameters in RF. The PSO algorithm was implemented in Python using the PyCharm2020.1.5 software , the libraries of which greatly facilitated our work. Firstly, a group of random particles is generated as the initial population. Each particle includes a set of hyperparameters for the random forest (n_estimators and max_depth) and has an initial position and velocity. Then, the accuracy obtained from ten-fold cross-validation is used as the fitness value for each particle. Combining the best position of the particle itself (individual optimal solution) and the best position of the entire population (global optimal solution), the velocity and position of the particle are updated. Finally, the model undergoes multiple iterations until the convergence conditions are met. The output is a random forest model with the optimal combination of hyperparameters.

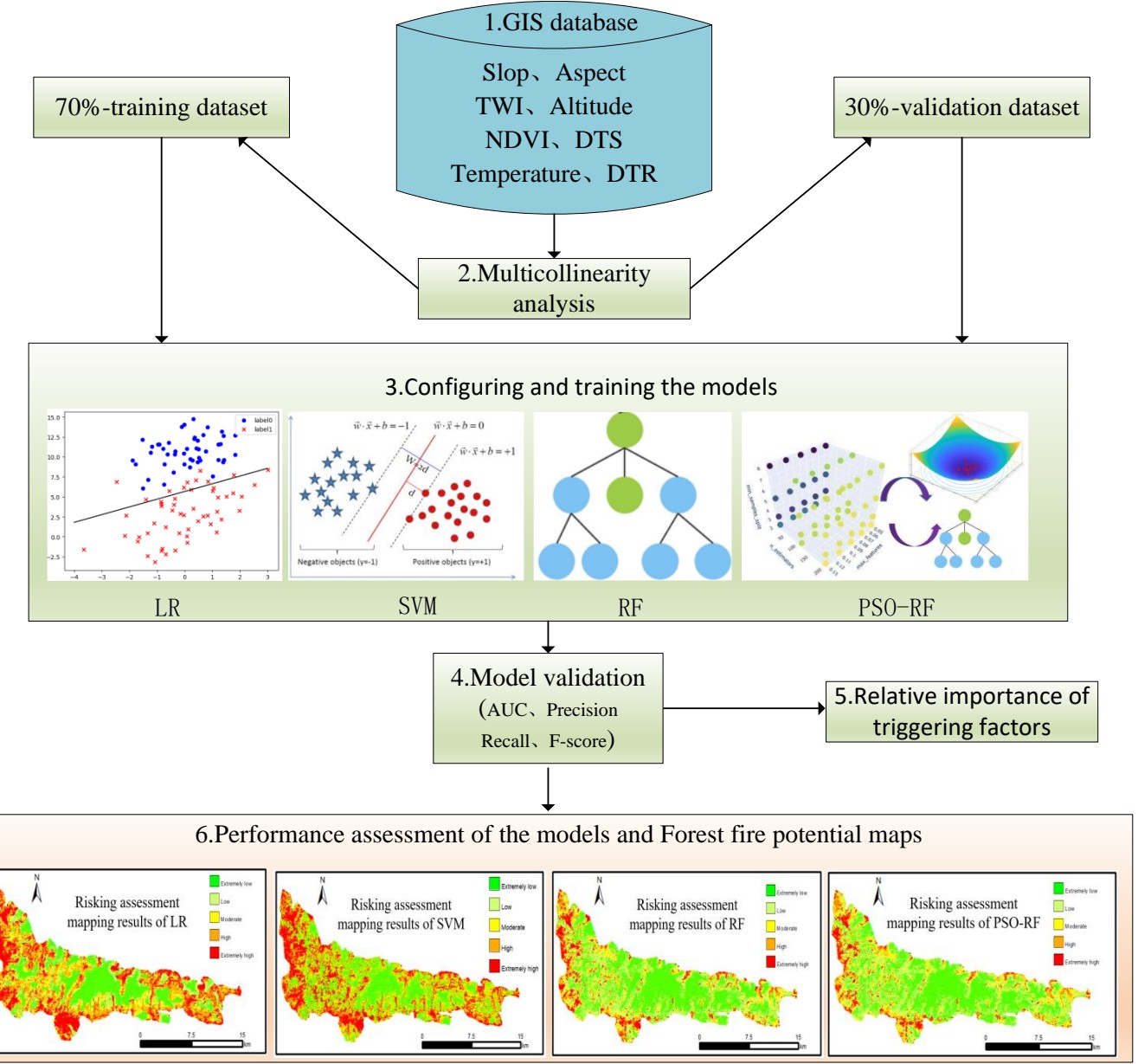

**Figure 3.** The general workflow shows the interaction from data input to the produced fire susceptibility map.

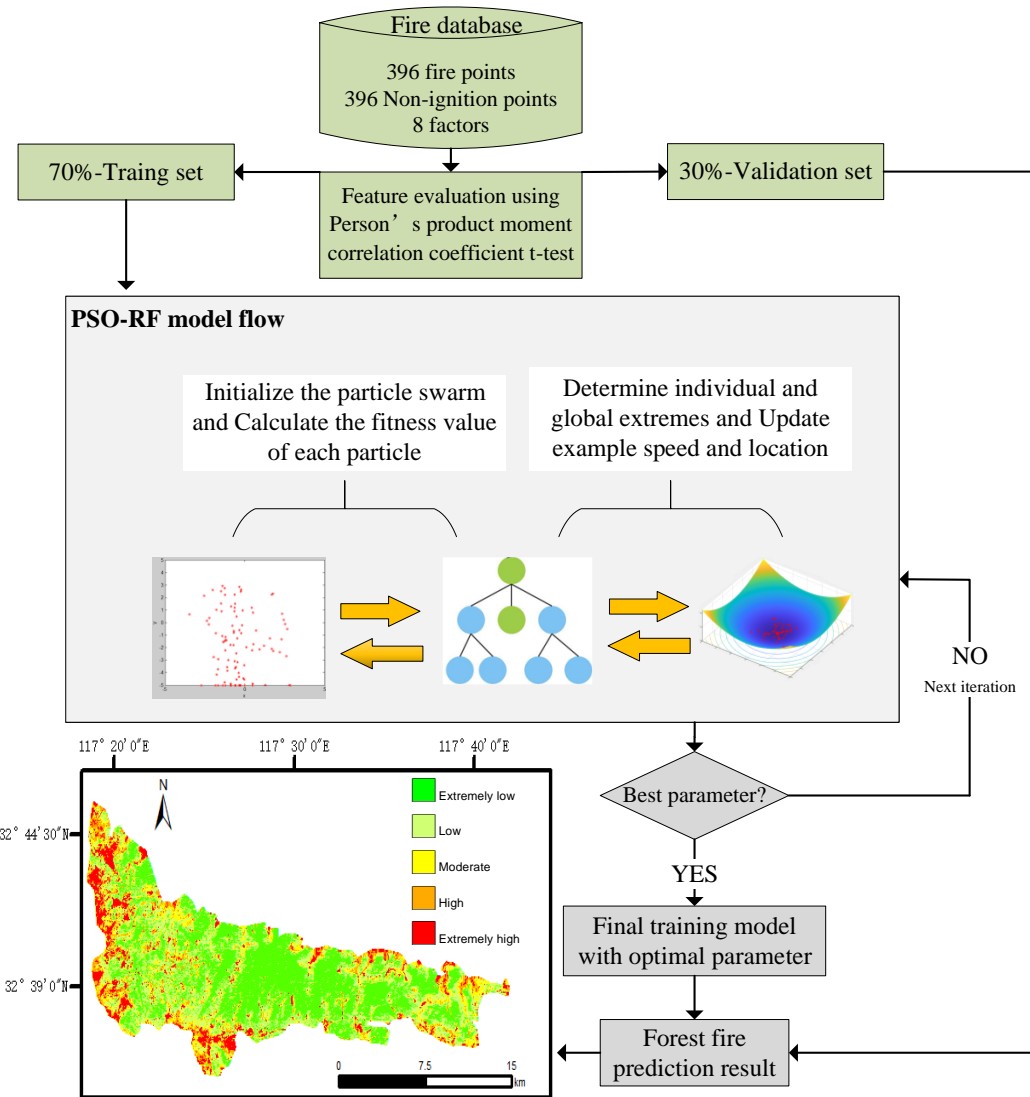

**Figure 4.** Flowchart of the PSO-RF model.

## 3. Results

### 3.1. Correlation Analysis of Variables

In this research, we conducted a multicollinearity diagnosis for all forest fire driving factors. By calculating the VIF values of each factor, Table 2 showed that the VIF values of all 8 factors were less than 5, indicating no multicollinearity among the factors [47]. Thus, there was no need to eliminate any influencing factors in this study.

**Table 2.** Multicollinearity analysis for relevant factors.

| Relevant Factors | VIF | TOL |
|---|---|---|
| Slope | 4.1 | 0.24 |
| Aspect | 3.1 | 0.32 |
| TWI | 1.8 | 0.56 |
| Altitude | 2.7 | 0.37 |
| NDVI | 1.1 | 0.91 |
| DTR | 2.4 | 0.42 |
| DTP | 3.6 | 0.28 |
| MMT | 3.5 | 0.29 |

To identify the key driving factors closely related to fire occurrence, this study chose the Pearson correlation coefficient method to conduct a correlation analysis between various variables and fire points [48,49]. Table 3 visually shows the correlation between the eight forest fire influencing variables and the fire point data. As evident from Table 3, all 8 factors exhibit P-values below 0.01, indicating substantial correlations among them. The highest degree of correlation is between DTR and NDVI, while the lowest degree of correlation is between the TWI and aspect.

**Table 3.** The relationship between the independent variables and dependent variables.

| Relevant Factors | Correlation Coefficient | *p*-Value |
|---|---|---|
| Slope | 0.26 | $p < 0.01$ |
| Aspect | 0.04 | $p < 0.01$ |
| TWI | 0.12 | $p < 0.01$ |
| Altitude | 0.3 | $p < 0.01$ |
| NDVI | 0.36 | $p < 0.01$ |
| DTR | 0.39 | $p < 0.01$ |
| DTP | 0.18 | $p < 0.01$ |
| MMT | 0.28 | $p < 0.01$ |

*3.2. Model Performance Evaluation*

We constructed a confusion matrix from the prediction results and evaluated the model performance using precision, recall, F-value, and AUC from the classification model evaluation metrics. The AUC value under the ROC curve served as the final evaluation metric for the model [48]. A larger AUC value signifies better classification performance. When the AUC value is greater than 0.9, it is considered excellent classification [50]. The relevant formulas are as follows:

$$Precision = \frac{TP}{TP + FP} \tag{3}$$

$$Recall = \frac{TP}{TP + FN} \tag{4}$$

$$F1\text{-}score = \frac{2 * Precision * Recall}{Precision + Recall} \tag{5}$$

$$AUC = \frac{1}{2}\left(\frac{TP}{TP + FN} + \frac{TN}{TN + FP}\right) \tag{6}$$

Our primary focus revolved around evaluating the model's performance on the test set. From Table 4, it can be observed that the AUC value of PSO-RF (0.908) is the highest, followed by RF (0.877), SVM (0.876), and LR (0.846). The AUC values of all these models surpass 0.8, making them widely employed algorithms in forest fire prediction research. The ensemble model (PSO-RF) proposed in this paper outperforms the other three models in terms of accuracy and generalization ability, making significant improvements over the random forest (RF) model based on experience and subjective assignment.

**Table 4.** Forest fire driver correlations.

| Model | Sample Type | TP | TN | FP | FN | AUC | Precision | Recall | F Value |
|---|---|---|---|---|---|---|---|---|---|
| LR | Training set | 210 | 222 | 51 | 70 | 0.851 | 0.805 | 0.75 | 0.776 |
|  | Validation set | 93 | 91 | 32 | 22 | 0.846 | 0.744 | 0.809 | 0.775 |
| SVM | Training set | 241 | 248 | 31 | 34 | 0.933 | 0.886 | 0.876 | 0.881 |
|  | Validation set | 104 | 87 | 30 | 17 | 0.876 | 0.776 | 0.860 | 0.816 |
| RF | Training set | 275 | 259 | 11 | 9 | 0.999 | 0.962 | 0.968 | 0.965 |
|  | Validation set | 94 | 100 | 26 | 18 | 0.877 | 0.783 | 0.839 | 0.810 |
| PSO-RF | Training set | 262 | 271 | 7 | 13 | 0.999 | 0.974 | 0.953 | 0.963 |
|  | Validation set | 104 | 98 | 20 | 16 | **0.908** | **0.839** | **0.867** | **0.852** |

### 3.3. Prediction Results of Fire Risk Map Level

Each pixel's fire occurrence probability within the research zone was classified using the natural break approach through ArcGIS 10.2 software. The natural break method of classification is based on the distribution characteristics of the data, determining classification boundaries according to the natural gaps between data points. Compared with the threshold method, which is subjectively set, the natural break method is more objective and reliable, and it is suitable for delineating forest fire risk levels. This is consistent with previous research [42]. Fire point probabilities were divided into five categories: very low, low, medium, high, and very high [51]. Figure 5 presents the fire risk maps drawn by the four models.

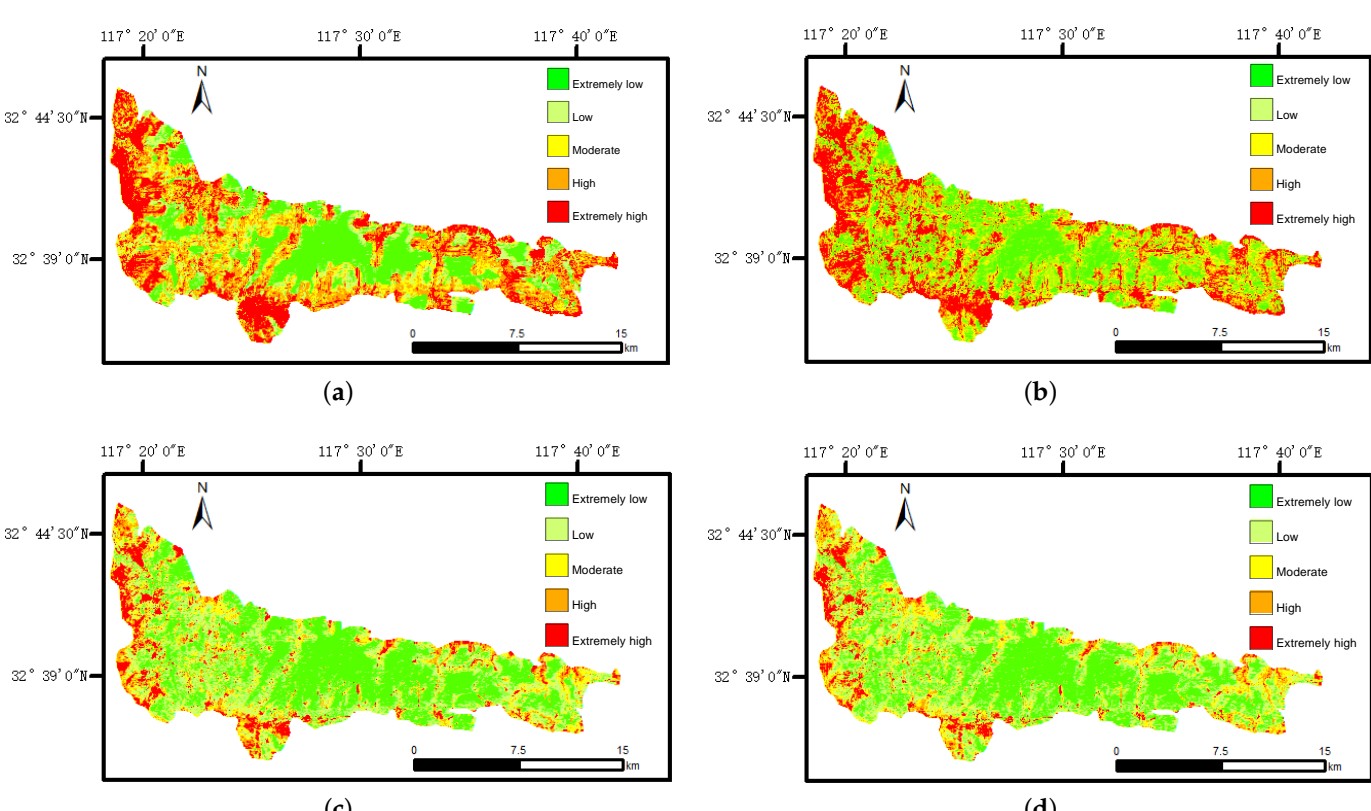

**Figure 5.** Forest fire susceptibility maps. (**a**) LR, (**b**) SVM, (**c**) RF, and (**d**) PSO-RF.

In this study, four machine learning models (LR, RF, SVM, and PSO-RF) were compared. Firstly, the LR model had the lowest predictive performance, which may be due to its lower performance in discrete and irregular data than non-parametric models (SVM and RF). In various indicators of the training and validation sets, the LR model performed poorly. After classifying the fire probabilities of each grid using the natural break method,

38.43% belonged to the very low- and low-risk areas, 16.24% to the medium risk area, and 45.33% to the high-risk area.

Secondly, the predictive performance of the RF model was superior to LR and SVM but inferior to the PSO-RF model. After classification by the natural break method, 66.69% belonged to very low- and low-risk areas, 16.3% to medium-risk areas, and 17.01% to high-risk areas. The predictive performance of the SVM model was only higher than LR but lower than the other two models. After classification by the natural break method, 46.96% of the area was classified as very low and low risk, 16.11% was classified as medium risk, and 36.93% was classified as high risk.

Finally, the PSO-RF model had the greatest predictive performance. The credibility of the drawn fire risk map was the highest, serving as an important reference for the forest fire work in the Jiushan area. In the risk map, 70.13% belonged to the very low- and low-risk areas, 14.03% to medium-risk areas, and 15.23% to high-risk areas. Combining Figures 5d and 2a,f, we found that the red areas (very high) were densely distributed at the foot of the mountain and on both sides of the road.

Figure 6 shows the proportions of these four models in the distribution of each fire risk level. We designated areas with very low and low fire-risk levels as regular management areas, and those with high and very high fire-risk levels as alert areas. The proportions of regular management areas for the LR and SVM models were below 50%, which is obviously not reasonable given that the average number of fire points extracted per year is 132, suggesting that most areas should be regular management areas. The proportion of regular areas in the PSO-RF model was 70.73%, which clearly aligns with reality.

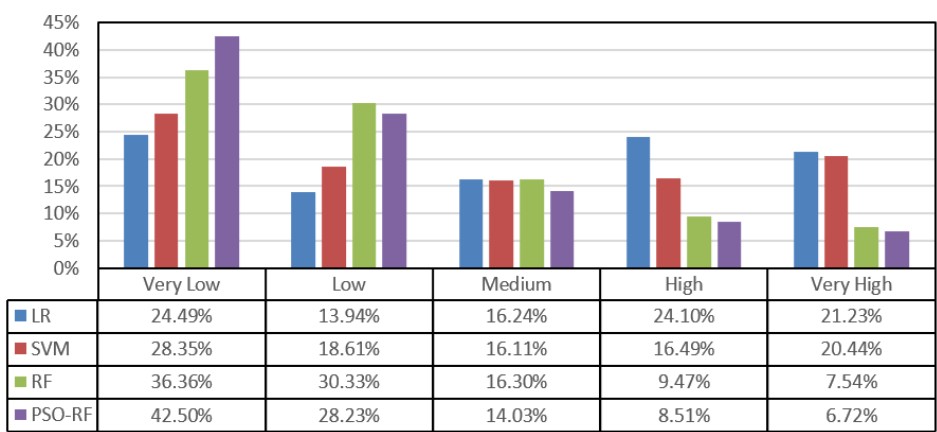

**Figure 6.** Specific rank distribution of the fire risk map.

### 3.4. Importance Evaluation of Influencing Factors

In this study, we selected eight factors influencing fires based on the experience of previous researchers (slope, aspect, TWI, altitude, NDVI, DTR, DTP, and MMT) as inputs for the models (LR, SVM, RF, and PSO-RF). Finally, we drew a fire sensitivity map for the study area. To further explore the driving factors of forest fires in the Jiushan area and measure the relative importance of input features in forest fire data, we used the feature importance evaluation method in machine learning models [52].

The analysis of model performance demonstrated that PSO-RF exhibited superior performance among the four models, indicating its ability to effectively elucidate the correlation between fire points and various triggering factors. We used the PSO-RF model for feature importance evaluation. The results showed that among the factors causing fires, the NDVI has the highest importance, followed by Altitude and DTR. Figure 7 specifically shows the importance of each triggering factor.

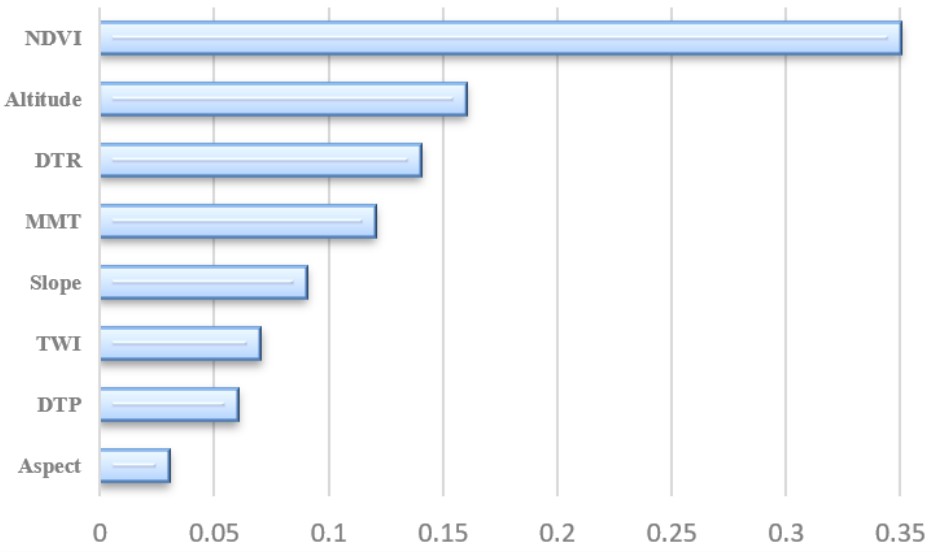

**Figure 7.** Ranking the importance of triggering factor features.

## 4. Discussion

This study uses the Jiushan area as an example, using the PSO-RF algorithm to establish a forest fire prediction model. Standard machine learning frameworks like LR, SVM, and RF were selected for comparative analysis. The assessment of these models revealed that PSO-RF topped the list with an AUC value of 0.908, succeeded by RF (0.877), SVM (0.876), and LR (0.846). The findings reveal that the accuracy of the RF model was enhanced post-optimization with the PSO algorithm. This facilitated the creation of a more precise fire risk map for the area under study, thereby offering more scientifically informed guidance for managing departments. We selected eight factors influencing fires as inputs for the model: slope, aspect, TWI, altitude, NDVI, DTR, DTP, and MMT. After multicollinearity diagnosis, there is no multicollinearity relationship among these eight factors, and they can be used to build a forest fire prediction model. A Pearson correlation analysis was conducted between these factors and fire point data, and the results show that the factors with a higher correlation with fire points are DTR, NDVI, altitude, and MMT. The feature importance results of the PSO-RF model showed that NDVI was the most important factor in causing fires, a conclusion also drawn by related scholars [21,53], followed by altitude and DTR. The most important factors causing forest fires can vary due to the differences in the study area. In his research on the Pu Mat National Park area in Vietnam, Tran [42] identified DTP as a key influencing factor, finding that fires in the area were primarily caused by human carelessness. Meanwhile, Mario discovered that DTR was a critical driver of forest fires in the southern region of Italy [54]. Human activity has been recognized as an important trigger for forest fires [42,54,55].

NDVI reflects vegetation cover. Areas with higher NDVI usually have denser vegetation, which may cause fires to spread more easily [56]. The DTR reflects human activity; discarded cigarette butts, camping, traffic accidents, etc., can all potentially cause forest fires. The higher the temperature, the easier vegetation dries out, thus making it more flammable. Higher-altitude areas usually have lower temperatures and sparser vegetation, so the risk of fire may be relatively lower. The fire risk map drawn by the PSO-RF model shows that high-risk fire areas are densely distributed at the foot of the mountain and on both sides of the road. The altitude and slope of the foothill areas are lower, and there are more human activities on both sides of the roads. Table 3 also points out that DTR, NDVI, and altitude have a strong correlation with fire points. Therefore, the fire risk map drawn by the PSO-RF model has a certain reference significance. Evidently, local fire departments should pay attention to forest fire prevention at the foot of the mountains and on both sides of the roads.

In random forest models, the configuration of hyperparameters requires manual adjustment. For instance, the max_depth parameter determines the maximum depth of the decision tree. If the depth is too small, the model may suffer from underfitting, while excessively high depth may lead to overfitting. Thus, selecting appropriate hyperparameters is crucial for achieving optimal model performance.

Haoyuan Hong conducted an investigation into fire susceptibility in Dayu County, Jiangxi Province, China. Utilizing a genetic algorithm (GA), he identified the optimal mix of variables associated with forest fires and employed both RF and SVM to construct fire hazard maps. The results revealed the superior performance of the optimized RF model (AUC = 0.8495), which surpassed that of the original RF model (AUC = 0.8169) [53]. On a similar vein, Zohre embarked on an investigation in Iran's Minudasht Township, deploying boosted regression tree (BRT), generalized additive model (GAM), and RF to formulate a fire risk map. Here, the GAM proved to be the most efficient (AUC = 0.877), while the RF lagged behind (AUC = 0.7279) [57]. In other research conducted by Ngoc in the Thuan Chau area of Vietnam, the use of SVM, RF, and the perceptron neural network (MLP-Net) model was implemented in studying forest fire susceptibility. The MLP-Net model showed the highest predictive performance (AUC = 0.894), surpassing the RF model (AUC = 0.883) [41]. These studies indicate that relying solely on the RF model often leaves researchers unsatisfied, underscoring the need for optimization to improve its performance.

Past studies mainly relied on experience or used default parameters in ensemble packages to set the RF model's hyperparameters. However, due to the variability of fire point regional data, there is no standard uniform hyperparameter setting applicable to all situations. Although hyperparameters set based on experience or default settings can still achieve good results in general, adjusting hyperparameters to optimize model performance is still necessary for specific studies [58]. The method proposed in this paper, which uses the PSO algorithm to optimize the RF model's hyperparameters, further improves the model's accuracy and benefits our scientific research.

The RF model has multiple hyperparameters, and this study chose to optimize those that significantly impact the model using PSO. The PSO-RF model used in this study can obtain the optimal parameters within a limited parameter range. This method circumvents issues such as overfitting and computational resource squandering that are common with traditional approaches to hyperparameter configuration, thereby boosting the effectiveness of the RF model. However, when using PSO to optimize RF parameters in this study, the value of the fitness function each time was the result of a three-fold cross-validation of the RF model. As PSO needs multiple iterations to obtain the optimal parameters, the training time is relatively long.

## 5. Conclusions

Fire risk maps play a crucial role in forest fire risk management. Given the diverse terrains, climates, vegetation, and road systems across different study areas, the selection of a suitable methodology is crucial for the development of fire risk maps tailored to specific research regions. In the present research, the Jiushan region serves as the focal point, and a predictive model for forest fires is developed leveraging machine learning algorithms. We propose an ensemble model combining PSO and RF, with LR, SVM, and RF as comparison models. Historical fire point data obtained from Sentinel fire products was analyzed, and factors such as altitude, slope, aspect, TWI, DTP, DTR, NDVI, and MMT were used as model inputs. Model performance evaluation adopted confusion matrix indicators, ROC curves, and ten-fold cross-validation. A fire risk map of the Jiushan area was finally drawn.

Correlation analysis shows that the distance from fire points to roads in human activities and the NDVI in vegetation factors have a more significant impact on fire risk compared to other factors. The model evaluation results indicate that PSO-RF performed the best among the four models. Based on the feature importance analysis of the PSO-RF model, the primary triggers for wildfires in the Jiushan area are the NDVI, followed by altitude and DTR.

By combining the fire risk map drawn by the PSO-RF model, and integrating feature importance analysis and correlation analysis, we found that high-risk fire areas are primarily located near the foot of the mountains and on both sides of the roads. The temperature at the foot of the mountain is high, and the vegetation has a low water content, which can easily cause wildfires. Human activities are frequent in the foothills, which may lead to a higher amount of dead branches and leaves as frequent human activities could potentially cause vegetation to die. Vehicle traffic on the roads and surrounding human activities can also easily cause fires, for instance, cigarette butts discarded by drivers or open flames caused by vehicle breakdowns. Therefore, the conclusions of this study have practical value.

Fire prevention in the Jiushan area should focus on the daily management of forests, including the timely clearing of withered vegetation and the reasonable adjustment of vegetation density. Also, road management at the foot of the mountains needs to be enhanced, including the timely sweeping of flammable materials on the road, the strengthening of propagation, the education of forest fire prevention knowledge among tourists and local residents, and the improvement of safety awareness when using fire sources. Currently, forest fire prevention and control in the Jiushan area mainly rely on manual patrols and monitoring from lookout towers. Patrol routes are usually based on experience, and the monitoring range of lookout towers is limited, making fire prevention methods relatively outdated. Using fire risk maps to plan patrol routes, optimize the layout of lookout towers, and focus on deploying fire prevention resources in high-risk areas may become an effective fire prevention measure.

**Author Contributions:** Conceptualization, F.Z.; methodology, C.S.; software, C.S.; resources, F.Z.; data curation, C.S.; writing—original draft preparation, C.S.; writing—review and editing, F.Z.; visualization, C.S.; and supervision, F.Z. All authors have read and agreed to the published version of the manuscript.

**Funding:** This research received no external funding.

**Data Availability Statement:** The slope, aspect, TWI, and altitude data are derived from the SRTMDE-MUTM90M resolution digital elevation data in the DEM digital elevation dataset of the Geospatial Data Cloud platform (https://www.gscloud.cn, accessed on 10 July 2021). The distances between fire points and roads, as well as residential areas, were obtained from the 2017 data in the 1:250,000 National Basic Geographic Database available in the National Geographic Information Resources Catalogue Service System (https://www.webmap.cn, accessed on 10 July 2021). The Landsat-8 OLI data and Sentinel-2 satellite data can be downloaded from United States Geological Survey(USGS) (https://earthexplorer.usgs.gov, accessed on 10 July 2021).

**Conflicts of Interest:** The authors declare no conflict of interest.

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
