# Peer review of "A Forest Fire Susceptibility Modeling Approach Based on Integration Machine Learning Algorithm"

_forests, doi:10.3390/f14071506_

Round 1

Reviewer 2 Report

I extend my gratitude for extending the invitation. I have diligently scrutinized the manuscript at hand. The authors delve into the utilization of the Random Forest Classifier to derive a forest fire susceptibility map for a specifically chosen study area. Notably, the authors propose the application of Particle Swarm Optimization to fine-tune the hyperparameters of the Random Forest, presenting it as a fresh perspective. In my estimation, while the paper does not exhibit novelty in terms of methodology and material, it does possess a certain local novelty due to the production of the aforementioned map for the region, marking its inaugural creation. I have discerned no issues in terms of implementation, and it appears that the authors have endeavoured to present their input and output in an acceptable manner. The paper can be deemed a valuable contribution to the field, employing well-established techniques. Nevertheless, it lacks a thorough exploration of existing literature and displays some deficiencies in the presentation of the research's input and output. I shall now outline my concerns.

1) First and foremost, a significant stumbling block in the manuscript lies in the authors' failure to conduct an adequate literature review encompassing crucial debates. Stated differently, the authors have not truly addressed recent studies on forest fire susceptibility based on machine learning within the introduction (the cited resources lie outside the realm of susceptibility). Furthermore, the output of this research has not been comparatively discussed in relation to similar studies. To rectify this setback, the authors should thoroughly explore recent studies employing Random Forest and feature importance within the introduction, drawing comparisons between the significance of their factors and those found in other studies. This will serve to highlight the extent to which the authors' output corroborates, contradicts, or enhances the existing body of literature. I highly recommend incorporating highly-cited susceptibility studies utilizing machine learning techniques published since 2021. Here are a few examples:

10.1016/j.ecolind.2021.107869

10.1016/j.ecoinf.2022.101647

10.1016/j.ecoinf.2021.101397

10.1016/j.ecoinf.2021.101292

2) Secondly, with regard to the data, it is imperative that the authors specify the spatial resolution of each conditioning factor in the text, as well as the final resolution to which it was resampled. Additionally, the authors must indicate the time period for active fire pixels, NBR (Normalized Burn Ratio), and NDVI (Normalized Difference Vegetation Index), ensuring consistency across these variables. Furthermore, it is essential to mention the satellite sensor employed to generate the NDVI map. Lastly, I implore the authors to emphasize the relevance of active fire pixels by incorporating pertinent citations in the text, thus justifying the utilization of such data to geolocate the position of burn areas, especially considering the continued heavy reliance on official inventories within the literature.

3) Turning to the methodology, the authors should explicitly mention the train-test split ratio within the text, rather than solely presenting it in the figure. Additionally, they should clarify the rationale behind the random selection of non-fire (negative) samples in their data frame. Furthermore, the authors ought to specify the performance metrics employed for their machine learning models, preferably accompanied by relevant formulas. Lastly, it is crucial to identify the specific tool utilized for the implementation of the Particle Swarm Optimization (PSO) algorithm, whether it be Optuna or another platform.

4) In terms of the results, it would be advisable to include a table encompassing a multicollinearity analysis, displaying the Variance Inflation Factor (VIF) and Tolerance (TOL) values. Moreover, the correlation table (Table 2) lacks clarity. Are the authors employing a feature selection technique? If so, it would entail considering all cross-correlations for each pairing of features. If not, it is necessary to elucidate the purpose behind seeking correlations. Furthermore, in Table 3, I suggest highlighting the best performance for each column to underscore the results more effectively. Finally, please provide substantiation for the adoption of the natural breaks method to classify susceptibility in the final maps by incorporating relevant citations.

5) It has come to my attention that certain abbreviations are used prior to their complete elucidation. For instance, I discovered the expanded forms of DTR and DTP in the discussion section, despite their initial appearance in the materials section. Moreover, there appears to be an interchangeable usage of TWI and TMI within the text. It is imperative to review the consistent usage of abbreviations throughout the entirety of the manuscript.

6) Lastly, the legends accompanying all maps would benefit from enhanced visibility. The current format lacks legibility, and therefore necessitates improvement.

Best regards.

The authors may need to review certain punctuations and word choices.
